# Pairwise Matching of Intermediate Representations for Fine-grained Explainability

## Abstract

The differences between images belonging to fine-grained categories are often subtle and highly localized, and existing explainability techniques for deep learning models are often too diffuse to provide useful and interpretable explanations. We propose a new explainability method (PAIR-X) that leverages both intermediate model activations and backpropagated relevance scores to generate fine-grained, highly-localized pairwise visual explanations. We use animal and building re-identification (re-ID) as a primary case study of our method, and we demonstrate qualitatively improved results over a diverse set of explainability baselines on 35 public re-ID datasets. In interviews, animal re-ID experts found PAIR-X to be a meaningful improvement over existing baselines for deep model explainability, and suggested that its visualizations would be directly applicable to their work. We also propose a novel quantitative evaluation metric for our method, and demonstrate that PAIR-X visualizations appear more plausible for correct image matches than incorrect ones even when the model similarity score for the pairs is the same. By improving interpretability, PAIR-X enables humans to better distinguish correct and incorrect matches.[1]

## 1 Introduction

Similarity-based deep metric learning has proven to be highly effective for a variety of tasks, particularly fine-grained tasks including image retrieval (Wang et al., 2014), facial recognition (Hu et al., 2014; Schroff et al., 2015), and open-set categorization problems such as animal re-identification (re-ID) (Čermák et al., 2023; Schneider et al., 2018; Haurum et al., 2020; Andrew et al., 2021). However, for robust, trustworthy deployment of these systems, interpretability is key. Existing explainability techniques are often insufficient, producing coarse visualizations that do not adequately capture the fine-grained details important to many tasks (Achtibat et al., 2023). As shown in Figure 2, this makes it difficult to precisely interpret which factors contribute most to the predicted similarity between a given pair of images (Zhu et al., 2021).

One application where explainability for deep metric learning models on fine-grained images is *necessary* for trustworthy deployment is animal re-identification (re-ID)—the task of distinguishing individual members of a species. Animal re-ID is a crucial tool in ecology and conservation, serving as the foundation for key applications such as monitoring population trends and analyzing both individual and collective behaviors (Tan et al., 2022). It is an active area of machine learning research, and recent years have seen steady growth in the performance of deep vision models on this task (Schneider et al., 2018). Animal re-ID typically relies on subtle, highly-localized fine-grained features such as subtle variations in spot or stripe patterns, or contours of fins, flukes, or ears. In contrast, explainability techniques like Grad-CAM frequently highlight broad image regions relevant to identification across all individuals (*e.g.* highlighting the entire side of a giraffe, as in Figure 2), but fail to capture localized details that vary between individuals (*e.g.* the placement of specific patterns on the giraffe).

The demand for better explainability techniques for animal re-ID is driven by an ongoing shift from classical re-ID techniques, such as HotSpotter or CurvRank (Crall et al., 2013; Weideman et al., 2020), towards deep models that improve predictive accuracy and scalability to large datasets (Otarashvili et al., 2024; Čermák

---

[1]Our code is available at: `https://github.com/pairx-explains/pairx`

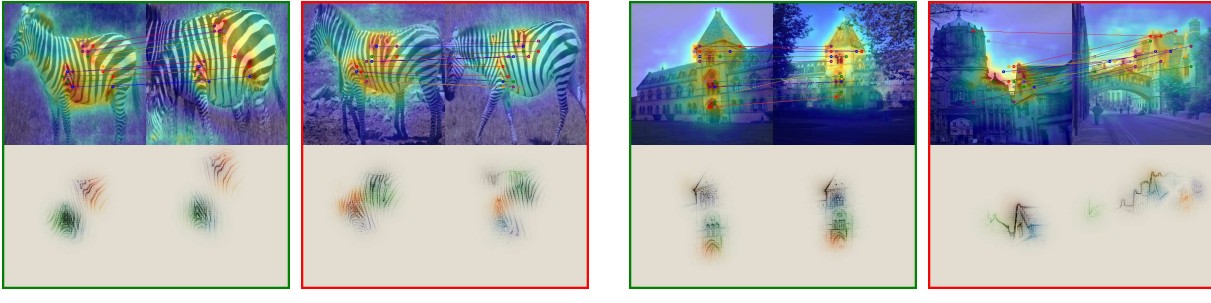

(a) Closest correct match  (b) Closest incorrect match  (c) Closest correct match  (d) Closest incorrect match

Figure 1: **PAIR-X provides interpretable, fine-grained, and highly-localized explanations which enable both correct and incorrect matches to be quickly identified.** The top half of each explanation shows pairwise-matched high-contribution deep features, and the bottom half shows a color-coded backpropagation to the original image pixels, highlighting plausible or implausible orientation shifts between fine-grained features.

et al., 2023; Schneider et al., 2018; Haurum et al., 2020; Andrew et al., 2021). Classical techniques for re-ID typically rely on algorithmic matching of localized features (*e.g.* SIFT (Lowe, 2004)), and are inherently explainable, as the local feature matches used for identification can be explicitly visualized. The resulting explanations can facilitate efficient manual review of model predictions, which is particularly necessary for high-profile applications such as population estimation for endangered species using visual mark-recapture, where incorrect labels can lead to significant errors in population size estimates (Stevick et al., 2001). In interviews with giraffe re-ID experts, we found that the shift from classical techniques to deep models had resulted in a significant loss of explainability, which made it more challenging and thus slower for experts to verify predictions.

To address this gap, we propose **P**airwise m**A**tching of **I**ntermediate **R**epresentations for e**X**plainability (**PAIR-X**), a post-processing method that produces fine-grained visual explanations for deep models that mimic and extend those of classical techniques, without retraining or architectural changes (see Figure 1). PAIR-X combines techniques from classic local feature matching with insights from modern explainability techniques for deep models (Bach et al., 2015; Zhu et al., 2021; Achtibat et al., 2023; Amir et al., 2022), and produces visualizations with the following key qualities:

- **Local pairwise correspondences.** PAIR-X mimics the explainability visualizations produced by classical feature matching techniques (as shown in Figure 2). Correspondences between local image regions can be explicitly visualized.

- **Fine-grained resolution.** PAIR-X produces explanations in the resolution of the original input image, thus capturing details in full resolution and shifting focus from broadly relevant regions to highly discriminative details.

- **Quantifiable metrics.** Previous explainability approaches have often relied solely on manual, qualitative review to measure performance. For PAIR-X, we additionally propose a set of quantitative metrics (see Section 3.3) to compare its performance across models and datasets. Our metrics are designed to approximate how plausible a given visualization will appear to a user.

We evaluate PAIR-X across 34 public datasets for animal re-ID from WildlifeDatasets (Čermák et al., 2023) as well as the Oxford Building 5k dataset as a proof-of-concept outside of animal re-ID (Philbin et al., 2007). We find that PAIR-X performs best on fine-grained, pattern-based tasks, rather than cases where identity depends on more global or gestalt characteristics. Qualitatively, PAIR-X enables the visualization of interpretable local correspondences between image pairs, which are useful for both efficiently verifying correct matches and flagging high-scoring incorrect matches. We additionally propose a novel metric for our method which quantitatively demonstrates that PAIR-X produces measurably more plausible explanations for matching image pairs than for non-matching image pairs. PAIR-X can distinguish correct and incorrect pairs **even in some cases where the deep metric model fails** (*e.g.* high model match score for an incorrect pair).

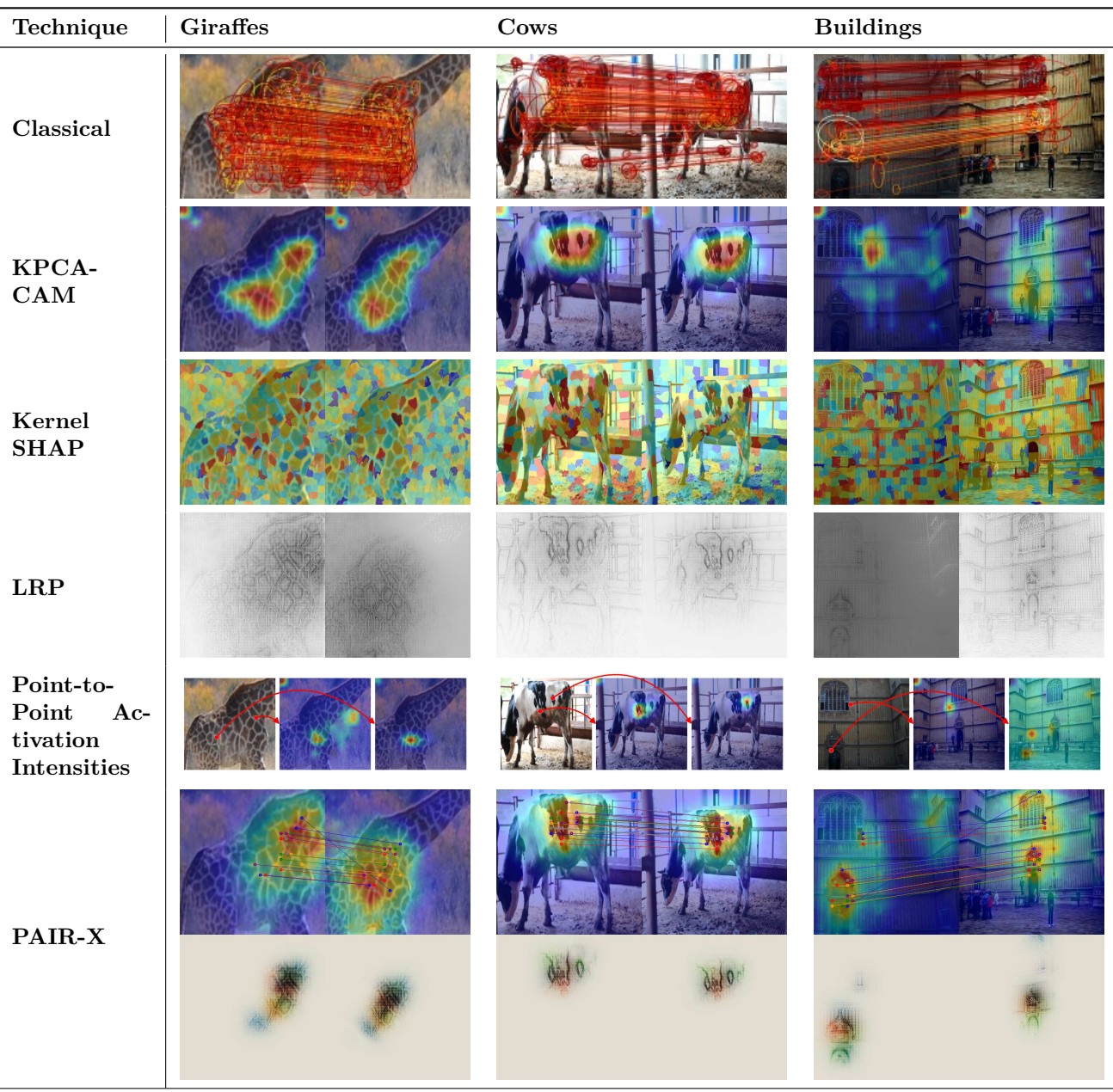

| Technique | Giraffes | Cows | Buildings |
|---|---|---|---|
| Classical | | | |
| KPCA-CAM | | | |
| Kernel SHAP | | | |
| LRP | | | |
| Point-to-Point Activation Intensities | | | |
| PAIR-X | | | |

Figure 2: Qualitative comparison of explainability techniques across image pairs selected randomly from three datasets. See Supplementary Figure 10 for a more expansive justification of baselines, including an ablation over 12 CAM-based techniques and a hyperparameter search for SHAP.

## 2 Related work

### 2.1 Animal re-ID

Historically, classical, inherently-explainable techniques based on homography-aligned local feature matching were used for individual re-ID of patterned (Crall et al., 2013; Lahiri et al., 2011; Kelly, 2001) and contoured species (Weideman et al., 2020; Hughes & Burghardt, 2017). Recently, deep models have been applied to a broad range of animal re-ID datasets (Otarashvili et al., 2024; Schneider et al., 2018; Čermák et al., 2023; Andrew et al., 2021; Haurum et al., 2020; Deb et al., 2018; Dlamini & Zyl, 2020). Deep metric learning

methods (Wang et al., 2014), including both convolutional neural network (CNN) (Otarashvili et al., 2024) and transformer architectures (Čermák et al., 2023), enable recognition of animals not seen during training, and have been shown to both generalize across species (Čermák et al., 2023), and scale more efficiently to large populations (Otarashvili et al., 2024). Concepts from local feature matching have been applied to deep vision model features for use in co-segmentation (Li et al., 2019b) and semantic correspondence (Ufer & Ommer, 2017). Like local features, these deep features can be matched between image pairs (Fischer et al., 2015; Amir et al., 2022; Balntas et al., 2017), but exploration into their use for explainability has been limited.

## 2.2 Explainability for deep vision models

Methods for generating saliency heat maps (*e.g.* , Grad-CAM, Grad-CAM++, DiffCAM, etc.) (Selvaraju et al., 2016; Chattopadhyay et al., 2017; Karmani et al., 2024; Draelos & Carin, 2021; Li et al., 2025; Zheng et al., 2020) are used to highlight image regions that contribute to the prediction of a certain class or to the similarity of features between pairs of images, but struggle to capture fine-grained features (Achtibat et al., 2023). Explainability methods which repeatedly perturb an image and measure the change in model output (Ribeiro et al., 2016; Lundberg & Lee, 2017) can be both computationally expensive and qualitatively ineffective for fine-grained details (see Fig. 2). Methods that build explainability into new model architectures can yield high-quality explanations, but require training specialized models from scratch (Chen et al., 2019). Layer-wise relevance propagation (LRP) backpropagates relevance back to the original image pixels and can capture fine-grained details (Bach et al., 2015). The initial relevance value is defined according to the model output, and relevance to that output is propagated backwards through the model according to a defined set of rules. This yields explanations in the spatial resolution of the original input; every image pixel can be assigned a relevance value. However, as seen in Figure 2, LRP fails to demonstrate how or why these details contribute to the model prediction. Concept relevance propagation (CRP) (Achtibat et al., 2023) combines the fine-grained, localized explanations of LRP with encoded, interpretable "concepts". While CRP can offer precise insights into a model's internal workings, it requires substantial human review to identify relevant and understandable concepts. To directly compare pairwise feature correspondences, Zhu et al. (2021) compute point-to-point activation intensity between pairs of images, and Nguyen et al. (2023) use optimal transport to match image regions. Both approaches rely on final-layer features, which are often coarse, and do not visualize receptive fields.

## 3 Our method

PAIR-X assumes access to a pretrained deep metric learning model (*i.e.* Kaya & Bilge (2019)) for the fine-grained task of interest, and, taking inspiration from classical feature-matching techniques, constructs an interpretable, highly localized post-hoc explanation of similarity between image pairs by combining intermediate deep feature matching (Fischer et al., 2015) and layerwise relevance propagation (Bach et al., 2015). These explanations can be generated for top-ranked pairs and used for human review and validation of matches. A visual description of our method can be found in Figure 3, and additional methodological details are captured in Suppl. Sec. C.

### 3.1 Deep feature matching

Local feature matching techniques typically operate on sets of keypoints $\mathcal{K}$ and descriptors $\mathcal{D}$, where keypoints describe spatial locations within an image, and descriptors provide information about features present at the keypoints. PAIR-X uses a simple spatial decomposition of an intermediate activation matrix to produce keypoint-descriptor sets $\mathcal{K}$ and $\mathcal{D}$ using the intermediate activations of a model (Fischer et al., 2015; Amir et al., 2022). Given an intermediate activation matrix $A^l$ at selected layer $l$ of shape $w \times h \times c$, which can be viewed as a spatial grid of $c$-dimensional feature vectors, where $w \times h$ represents the spatial resolution at layer $l$, we decompose $A^l$ into $N = w \times h$ descriptors of length $c$, thus $\mathcal{D} = \{A_{i,j}^l \mid i \in \{1, ..., w\}, j \in \{1, ..., h\}\}$. The keypoints $\mathcal{K}$ for each descriptor are simply defined according to the estimated location in the image, *i.e.* , $\mathcal{K} = \{(i,j) \mid i \in \{1, ..., w\}, j \in \{1, ..., h\}\}$. As this definition of keypoint locations does not fully capture

the true receptive fields for each neuron, we additionally utilize LRP to create more precise visualizations (see Sec. 3.2). Given the complete sets of keypoints and descriptors, we perform brute-force matching with cross-checking, *i.e.* , descriptors $(x, y)$ will only be returned for a match if $x$ is the closest match to $y$, and vice versa. This yields a set of matches $M$.

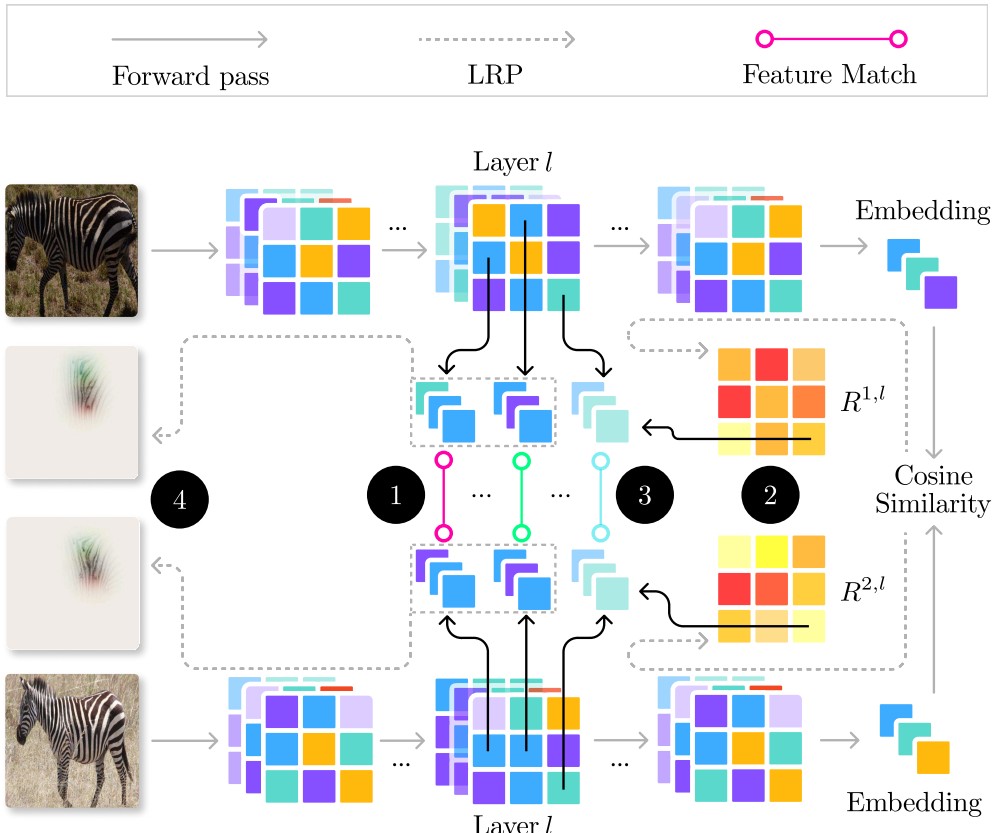

Figure 3: Overview of PAIR-X. In step ①, we match deep features derived from layer $l$. In step ②, we perform LRP to obtain the relevance of each feature to the final cosine similarity. In step ③, we filter the matched features acccording to their estimated relevance. Finally, in step ④, we use LRP to visualize which original image pixels are relevant to a filtered subset of matches.

## 3.2   Layerwise relevance propagation

Naive feature matching results in a large set of matches, some of which are unimportant to the model prediction. To filter this large set of candidate matches, we first use LRP to determine relevance values for each neuron in the selected intermediate layer $l$. The resulting matrix takes shape $w \times h \times c$, and represents the estimated relevances of the values in the intermediate feature map at $l$. Given a keypoint match between $(i_1, j_1)$ and $(i_2, j_2)$ in the first and second images, respectively, we compute a relevance score for the match:

$$Rel((i_1, j_1), (i_2, j_2)) = \left( \sum_k R^{1,l}_{i_1, j_1, k} \right) \times \left( \sum_k R^{2,l}_{i_2, j_2, k} \right) \tag{1}$$

where $R^{1,l}$ and $R^{2,l}$ are the intermediate relevance matrices for the two images, and we keep the $n$ highest-scoring matches ($n = 20$ in our figures).

Neurons typically draw information from a wider surrounding region, or receptive field, which is not captured by a visualization with lines connecting approximated keypoints. To precisely visualize the pixel-space

contributions of matched features, we use LRP to backpropagate from the selected intermediate layer $l$ to the original image for a set of top matches (ranked according to the relevance metric presented in Equation 1). Given a feature match between keypoints $(i_1, j_1)$ and $(i_2, j_2)$, we backpropagate for each image in the pair separately. We first mask the intermediate activation matrix, $A^l$, to the values included in the matched keypoint descriptor. This takes the form:

$$\mathcal{M}(A^l_{i,j}) = \begin{cases} A^l_{i,j}, & \text{if } (i,j) = (i_1, j_1) \\ 0, & \text{otherwise} \end{cases} \tag{2}$$

We then backpropagate relevance from $\mathcal{M}(A^l_{i,j})$, using the rules defined by LRP. The result of this backpropagation takes the same shape as the original input image, which can be summed channel-wise for RGB images to produce a 2D heatmap. We visualize the pixel-wise relevances by assigning a different color map to each match, then combining all matches into a single color-coded visualization.

### 3.3 Quantitative explainability metrics

Because explainability is inherently qualitative, it is difficult to define metrics that can quantify explainability performance. We propose two quantitative metrics which seek to capture the "plausibility" of PAIR-X explanations.

**Inverted residual mean.** Our first metric, the inverted residual mean, aims to capture whether the feature matches follow a "ground truth" homography $\mathcal{H}$, as a proxy to understand whether the matches correctly align the image subject. Matches that do not follow a homography will typically appear less plausible. $\mathcal{H}$ is calculated using classical techniques; we use a SuperPoint extractor to extract local features and a LightGlue matcher to find feature matches, then estimate $\mathcal{H}$ from the matches (DeTone et al., 2017; Lindenberger et al., 2023). Each keypoint $p_1 = (i_1, j_1)$ is projected from the first image using this "ground truth" homography as:

$$\mathcal{H}\left(\begin{bmatrix} i_1 \\ j_1 \\ 1 \end{bmatrix}\right) = \begin{bmatrix} i'_1 \\ j'_1 \\ w' \end{bmatrix} \rightarrow p'_1 = \frac{1}{w'} \begin{bmatrix} i'_1 \\ j'_1 \end{bmatrix}. \tag{3}$$

For the final score $S_1$, we take the reciprocal of the average of the residuals on the second image (between the projected points and the matched points) across all feature matches $M$ (before LRP filtering):

$$S_1 = \frac{|M|}{\sum_{(p_1, p_2) \in M} ||p'_1 - p_2||}. \tag{4}$$

We take the reciprocal because residual means are typically concentrated around small values, with a long tail of high-value outliers. Taking the reciprocal allows for improved separability of the smaller values.

**Relevance-weighted match coverage.** The second metric we propose, relevance-weighted match coverage, aims to understand what proportion of relevant regions are successfully matched by PAIR-X. Visualizations that fail to show matches between the most important regions of an image will also appear less informative. Each feature that has been matched by PAIR-X is weighted by its relevance score, summed, and then divided by the sum of all relevance scores:

$$\frac{\sum_{(i,j) \in \mathcal{K}_1} R^1_{i,j} + \sum_{(i,j) \in \mathcal{K}_2} R^2_{i,j}}{\sum_{i,j} R^1_{i,j} + \sum_{i,j} R^2_{i,j}}, \tag{5}$$

where $\mathcal{K}_1$ and $\mathcal{K}_2$ denote the sets of matched keypoints of the two images, before being filtered by relevance. For each dataset evaluated, we compute these metrics across top-ranked correct and incorrect pairs following the procedure described in Suppl. Sec. C.4.

**Patterned species**

PAIR-X performs best on fine-grained tasks with highly-localized and structured details, e.g. re-ID for patterned species such as giraffes, cows, zebras, and sea turtles. These patterns capture unique spatial arrangements for individuals, thus the visualizations for correct and incorrect matches are interpretably different. Green outlines indicate correct image matches (same individual), and red outlines indicate incorrect matches (different individuals).

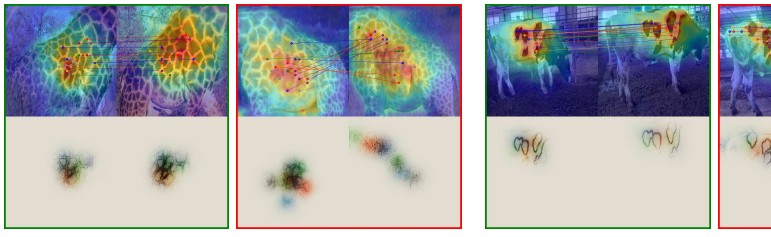

Giraffes - Correct and Incorrect Matches      Cows - Correct and Incorrect Matches

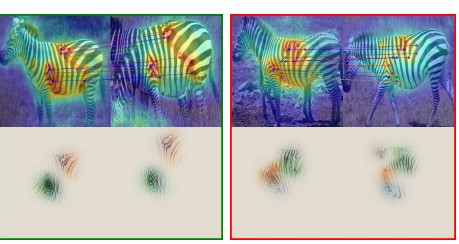      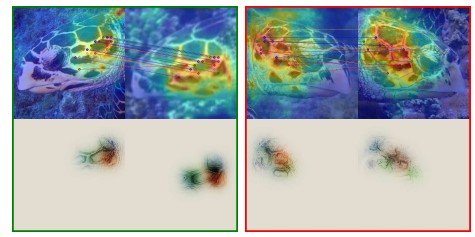

Plains Zebras - Correct and Incorrect Matches      Sea Turtles - Correct and Incorrect Matches

**Unpatterned species**

PAIR-X is designed to optimally explain fine-grained, localized features that follow unique spatial arrangements for different categories. **For species without structured biometric patterns such as stripes, we find PAIR-X explanations are more useful for individuals with unique markings.** Without these markings, PAIR-X can sometimes highlight features that are invariant between individuals, producing misleading explanations for incorrect matches.

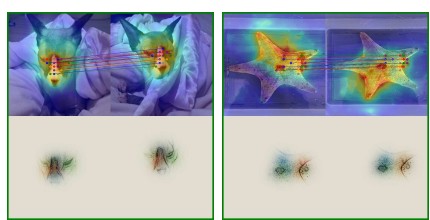

Cat and Sea Star Images with Distinctive Markings

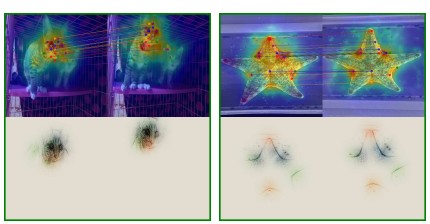

Cat and Sea Star Images without Distinctive Markings

**Identifying spurious correpondences**

PAIR-X can help identify when model decisions are based on irrelevant information. To the right, it shows the model's focus on foreground information.

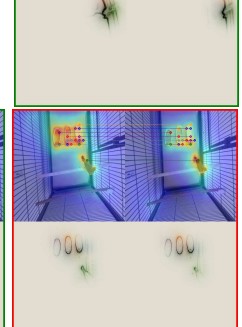

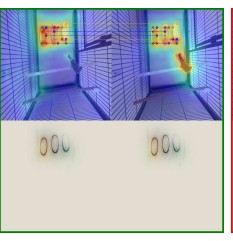

For these bird images, PAIR-X captures the background information contributing to the model prediction.

**Failure mode: extreme pose variation**

As with many classical feature-matching techniques, PAIR-X performance degrades as pose variation becomes increasingly extreme.

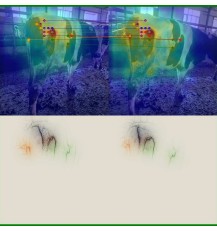      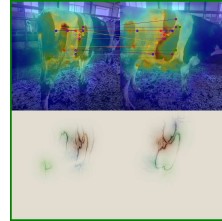      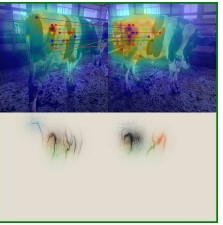      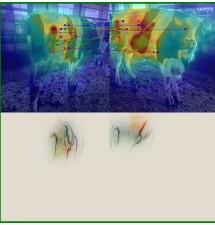

Minor pose difference      Moderate pose difference      Major pose difference      Extreme pose difference

Figure 4: Qualitative analysis of trends in PAIR-X outputs.

| Dataset | $\rho_{\mathrm{res}}$ | $\Delta_{\mathrm{res}} \uparrow$ | $\rho_{\mathrm{mc}}$ | $\Delta_{\mathrm{mc}} \uparrow$ |
|---|---|---|---|---|
| Cows2021v2 (Gao et al., 2021) | 0.77 | 0.74 | 0.80 | 1.32 |
| **Giraffes** (Miele et al., 2021) | **0.74** | **0.57** | **0.75** | **0.85** |
| Oxford5k (Philbin et al., 2007) | 0.64 | 0.37 | 0.69 | 0.58 |
| **DogFaceNet** (Mougeot et al., 2019) | **0.48** | **-0.01** | **0.70** | **0.02** |
| SMALST (Zuffi et al., 2019) | 0.46 | $-0.48$ | 0.67 | $-0.19$ |
| Average (35 datasets) | 0.50 | 0.18 | 0.64 | 0.23 |

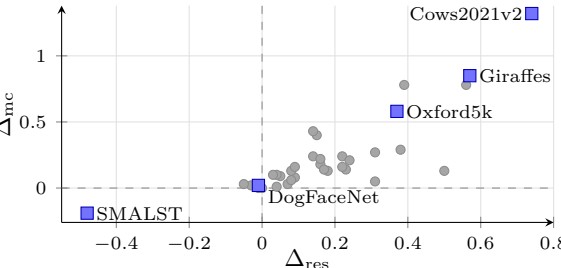

Figure 5: Quantitative metrics across datasets. Left: Inverted residual mean (*res*) and relevance-weighted match coverage (*mc*) across five representative datasets, aggregated within each dataset using both Spearman's rank correlation coefficient ($\rho$) between each metric and model match score, and binned Bhattacharyya distance ($\Delta$) of each metric between correct and incorrect matches. Positive $\Delta$ values indicate that PAIR-X improves separation for similarly scored matches. Metrics for **bolded** datasets are shown in more detail in Figure 6. Right: each point represents one dataset, plotted by the separability terms $\Delta_{\mathrm{res}}$ and $\Delta_{\mathrm{mc}}$. Full results across all datasets can be found in Suppl. Table 1.

## 4  Results

Using the multispecies re-ID model Miew-ID[2] as our deep metric model (Otarashvili et al., 2024), we evaluate PAIR-X across 34 public datasets for animal re-ID from WildlifeDatasets (Čermák et al., 2023), as well as the Oxford5k building dataset (Philbin et al., 2007). Our evaluation has two parts: (i) qualitative comparison of PAIR-X against diverse baseline explainability methods (Figure 10, Supplemental Figure 2), and (ii) quantitative evaluation of PAIR-X itself using the method-specific metrics introduced in Section 3.3. See Suppl. Sec. E for results on an additional model, and Suppl. Sec. H for a preliminary expansion to transformers. The metrics defined in Sec. 3.3 (which we refer to as the PAIR-X metrics) are aggregated across each dataset using two additional values described below. Results are presented in Figure 5. In Figure 6, we visualize our metrics across individual image pairs in specific datasets.

**Dataset Metrics.** Both metrics are computed for a fixed set of image pairs for each dataset (see Suppl. Sec. C for details on pair selection). To aggregate these metrics across pairs within each dataset, we computed two additional values for each PAIR-X metric. First, we measure the rank correlation between the PAIR-X metric and the model similarity scores, to ascertain whether PAIR-X visualizations appear more plausible for pairs that the model scores more highly. This is done using Spearman's rank correlation coefficient $\rho$ (see Figure 5). Second, we measure the separability of correct and incorrect pairs using the PAIR-X metric, while controlling for the model similarity score. The goal of this separability test is to ascertain whether, for correct and incorrect pairs that the model cannot distinguish, PAIR-X visualizations appear more plausible for correct matches than incorrect ones. This value is denoted as $\Delta$ in Figure 5, and it is measured by binning over cosine similarity, then taking a weighted average of bin-wise Bhattacharyya distances (see Suppl. Sec. C.3 for exact details). In Figure 6, we visualize and discuss possible distributions of our metrics for correct and incorrect pairs, to develop intuition for the meaning of these metrics.

## 5  Discussion

### 5.1  Performance across applications

**PAIR-X performs best on fine-grained tasks with highly-patterned or highly-localized features.** As an example, the giraffe data in WildlifeDatasets consists of high-quality, well-cropped images of Reticulated Giraffes, a species with dense, uniquely oriented patterns of highly-localized features. As shown in Fig. 6, the PAIR-X scores for this dataset follow two relevant trends. First, we see a strong positive correlation between match score and PAIR-X score, indicating that PAIR-X visualizations are more plausible for

---

[2]weights publicly available at `https://huggingface.co/conservationxlabs/miewid-msv2`

higher-scoring image pairs. It also suggests that PAIR-X is unlikely to produce highly plausible but misleading visualizations for image pairs with low model match scores. Second, we see that the PAIR-X scores show an additional dimension of separability between correct and incorrect pairs. For image pairs that the model assigns equivalent match scores, the PAIR-X metrics suggest that visualizations appear, on average, more plausible for correct than incorrect pairs. This is a promising result, and if additional separability between correct and incorrect pairs can be achieved through this type of method, it could perhaps be directly utilized to improve model accuracy.

Our method shows potential for fine-grained explainability beyond animals, particularly for other tasks with highly-structured and localized features such as building facades. We conduct a detailed analysis of performance on the Oxford5k dataset in Suppl. Sec. D.1.

**PAIR-X is less well-suited for fine-grained tasks with less-localized or less-structured distinguishing features**. As an example, for images in DogFaceNet, we see a much lower degree of separability between correct and incorrect pairs. Qualitatively, we see that this is largely due to structural similarities between individuals: PAIR-X is, for instance, likely to find matches between eyes and noses of incorrectly-matched dog pairs, especially when comparing different individuals from the same breed. This raises the question of what optimal explainability would look like in this case, where identification may be more gestalt than localizeable (*i.e.* subtle variations in relative spacing between eyes and nose, as opposed to unique patterns of stripes). As we see in Figure 6, many of the incorrect pairs that receive high model similarity scores are of very similar-looking dogs. In these cases, PAIR-X provides informative visualizations of the features that contribute most to similarity (*e.g.* eyes, nose). However, as these facial features contribute to similarity for both correct and incorrect matches, PAIR-X may produce misleading visualizations for incorrect matches, and is thus not as useful for manual review of model predictions. This is in contrast to highly patterned species, where highly-matched but uniquely-structured patterns for each individual are easily distinguished in our visualizations, and thus lead to higher PAIR-X metric scores. Examples on additional less-patterned species (cats and starfish) are shown in Figure 4.

## 5.2 Quantitative comparisons between methods

Ideally it would be possible to not only quantify the performance of our method across datasets, but also quantitatively compare our method to other explainability techniques. General-purpose quantifiable metrics for explainability are still out of reach, and our PAIR-X metrics assume explicit pairwise feature matching to calculate, and are thus not directly applicable to, e.g., CAM-based methods. Thus, we rely on qualitative comparisons and expert interviews to measure the relative interpretability and usefulness of explanations from different methods. That said, we want to highlight the value that our method-specific metrics provide. The ability to quantify explainability in PAIR-X allows a user to efficiently determine whether PAIR-X is a good fit for their task of interest.

## 5.3 Applicability to real-world use cases

In an envisioned use case for animal re-ID, PAIR-X visualizations could be used to more efficiently manually validate model predictions, especially in cases where the closest correct match and closest incorrect match are scored similarly. Qualitatively, PAIR-X visualizations help to isolate important information and to visually align images, reducing the manual labor required for match verification. As discussed in Section 4, the difference in PAIR-X metrics between correct and incorrect pairs with similar match scores suggests that visualizations for correct pairs would, on average, appear more visually plausible. This is especially true for datasets with a high degree of separability, as measured by $\Delta$ in Figure 5. We further analyze the applicability of PAIR-X to real-world use cases via expert interviews and a brief analysis of computational costs.

### 5.3.1 Expert interviews

Because explainability is highly subjective, we found it important to consider perspectives from downstream model users about the usability of PAIR-X. Animal re-ID is a niche topic, with a very limited number of experts capable of manual re-ID on these datasets, which limited the pool of people from whom to collect

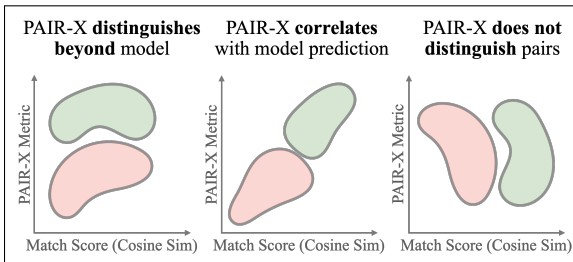

Three possible distributions of correct and incorrect matches using our PAIR-X metrics.

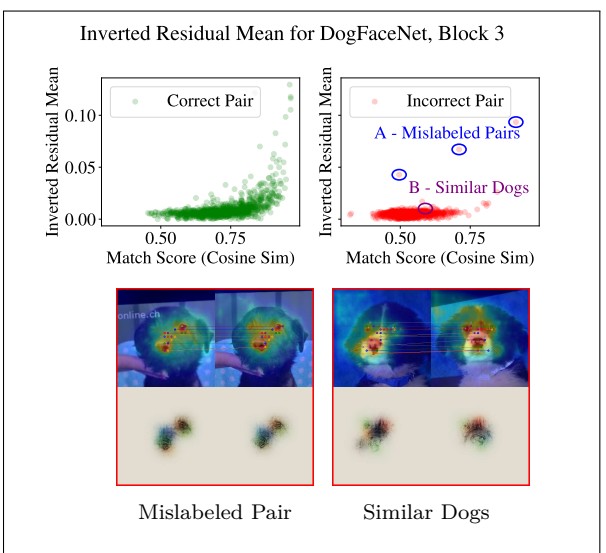

Mislabeled Pair          Similar Dogs

The PAIR-X metric struggles to separate correct from incorrect matches for less-highly-patterned species like dogs. However, this analysis identified several outliers in the DogFaceNet dataset which were found to be mislabeled copies of the same images, with distortions such as rotations and cropping. This ability to flag mislabeled pairs is potentially useful. However, some incorrect pairs still receive moderately high scores from both the model and PAIR-X simply because they are similar-looking.

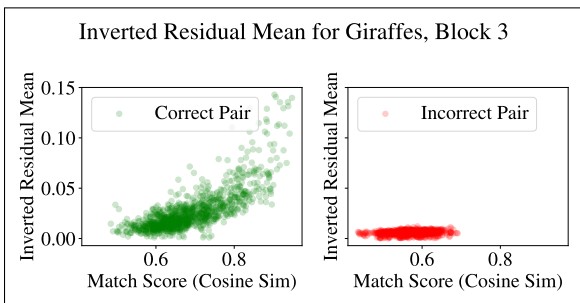

The PAIR-X metric separates correctly and incorrectly matched pairs well for highly-patterned species such as giraffes, suggesting that PAIR-X proves highly explainable in these cases.

Figure 6: We provide visualizations to build intuition for interpreting plots of PAIR-X metrics (inverted residual mean vs. match score), as well as real examples of these plots for a higher-performing (giraffe) and a lower-performing species (dogs).

feedback. However, we interviewed three experts in giraffe re-ID, and discuss a few key insights gained from those conversations. In real-world deployments of giraffe re-ID models, experts are tasked with manually verifying large batches of image labels, which frequently requires reviewing between five and twenty top-ranked database matches per query image. In this setting, efficiency is critical. The experts we interviewed agreed that explainability visualizations that highlight relevant image regions are helpful for directing user attention and speeding up verification.

We asked experts about their preferences between PAIR-X, SIFT feature matching such as HotSpotter, and Grad-CAM++. While experts found Grad-CAM++ to be more useful than *no* explainability, they found the fine-grained information provided by PAIR-X to be more helpful. Between classical SIFT feature matching and PAIR-X, experts were split. One expert had used HotSpotter extensively before switching to deep models, and thus had a preference for the classical feature-matching visualization, but recognized that its inability to scale to their current database rendered it no longer usable. Another expert, who had not previously used HotSpotter, found those visualizations to contain too many matches to be interpretable, and preferred PAIR-X for its filtered set of feature matches.

### 5.3.2 Computational efficiency

Since experts are interactively reviewing large numbers of images (one of the experts we interviewed had reviewed more than 100,000 during their time in the field), low latency, and therefore computational efficiency, is essential. On a single A100 GPU, we find that creating a typical explanation for 10 backpropagated matches requires 5 seconds, which demonstrates the feasibility of using PAIR-X at scale. We expand upon the factors influencing computational efficiency in Suppl. Sec. G.

## 6 Conclusion

We present PAIR-X, a novel fine-grained explainability technique based on a combination of deep feature matching and layer-wise relevance propagation (LRP), which provides explanations of pairwise similarity based on pretrained deep metric learning models. We demonstrate promising results on a diverse collection of animal re-ID datasets, as well as the Oxford-5k building dataset. Qualitatively, the explanations produced by PAIR-X are finer-grained than existing CAM-based techniques, as well as easier to interpret thanks to explicit matching of relevant image features and the color-coded propagation of those features back into image space. We furthermore propose a set of quantitative metrics which show that PAIR-X is in many cases able to distinguish correct from similarly scoring incorrect (*i.e.* confusing) matches. While PAIR-X may produce misleading explanations for species with a high degree of structural similarity, we show that there are many patterned species PAIR-X is applicable to. The experts we interviewed unanimously agreed that PAIR-X explanations are useful and informative, and WildMe[3], a cross-species animal re-identification platform, has expressed intent to deploy our method for all of its patterned species, emphasizing its applicability and usefulness in real-world settings.

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
