# OpenReview forum: "Pairwise Matching of Intermediate Representations for Fine-grained Explainability"
_TMLR — Rejected by TMLR_

### Review · Reviewer_qGV4 · 2026-03-02

**Summary Of Contributions:**

The authors proposed PAIR-X, a post-hoc explainability framework tailored for fine-grained visual recognition and re-identification tasks.
The main contribution is a correspondence-driven explanation mechanism that matches intermediate feature representations between image pairs to expose spatially localized evidence supporting similarity decisions.
By operating at the level of intermediate activations, the method provides structured insight into how identity-related cues are encoded inside deep networks.

In addition to the method, the paper introduces quantitative metrics for explanation evaluation.
A geometric consistency metric assesses residual alignment under estimated transformations, while a separability metric measures how well explanations differentiate correct from incorrect matches.
This moves beyond purely qualitative visualization toward reproducible, measurable evaluation.

The work is supported by large-scale experiments across more than thirty datasets, demonstrating strong performance in texture-rich, pattern-based re-identification domains.
The empirical design is systematic and includes failure case analysis and preliminary transformer experiments.

At the same time, the study reveals scoped limitations.
The method performs best when identity cues are spatially localized and structured, and its effectiveness diminishes in settings dominated by global or gestalt-based similarity.
The geometric metric relies on homography estimated via learned matchers, raising questions about proxy ground-truth assumptions.
Furthermore, transformer evaluation does not yet incorporate established transformer-specific relevance propagation methods, limiting claims of architectural generality.

Thus, the paper presents a technically coherent and empirically thorough contribution to correspondence-based explainability in fine-grained recognition, while leaving open important questions regarding metric robustness and architectural scope.

**Additional Comments:**

I do not have additional comments beyond the points raised above.

**Audience:**

Yes

**Audience Explanation:**

The findings of this paper would likely be of interest to a meaningful subset of the TMLR audience.
Explainable AI (XAI) remains a central research direction in machine learning, particularly as models are increasingly deployed in high-stakes and domain-sensitive applications.
While much of the XAI literature focuses on generic image classification benchmarks, fine-grained recognition and re-identification tasks present distinct interpretability challenges.
In such settings, decisions often depend on subtle, localized visual cues rather than coarse object-level features, and understanding correspondence structure between instances becomes particularly important.

The authors addressed this relatively underexplored intersection between fine-grained visual recognition and structured explainability.
Correspondence-based explanations at intermediate representation levels provide a complementary perspective to saliency or gradient-based methods, and may resonate with researchers working on metric learning, retrieval, representation analysis, and post-hoc interpretability.

Beyond methodological interest, the application to animal re-identification introduces practical relevance.
Wildlife monitoring, biodiversity conservation, and ecological research increasingly rely on automated identification systems.
In these contexts, interpretability is not merely aesthetic but functional: researchers and conservationists benefit from understanding which visual patterns drive identity decisions, especially when individual identification informs longitudinal studies or population management.

Thus, while the contribution may not target the entire TMLR readership, it sits at a meaningful intersection of explainability research, representation analysis, and applied computer vision in ecological domains.
For researchers interested in structured, correspondence-based explanations and their quantitative evaluation, the findings offer both methodological and practical insights.

**Broader Impact Concerns:**

I do not identify any significant broader impact concerns associated with this work.
The proposed method focuses on post-hoc explainability for fine-grained visual recognition and re-identification tasks, particularly in wildlife and ecological monitoring domains.
These applications are generally aligned with research transparency, scientific analysis, and conservation efforts.
The framework does not introduce new capabilities for surveillance, manipulation, or sensitive inference beyond standard computer vision techniques, and it does not rely on or target protected attributes.
Thus, the contribution appears technically focused and low-risk from a societal or ethical standpoint.

**Claims And Evidence:**

Yes

**Claims Explanation:**

The claims made in the submission are, within the scoped problem setting, supported by clear, systematic, and largely convincing empirical evidence.
The proposed PAIR-X framework is methodologically transparent: the correspondence-based matching mechanism at intermediate representation levels is clearly described, and its intuition--linking similarity decisions to spatially localized feature correspondences--is technically coherent and easy to interpret.

The evaluation metrics introduced in the paper further strengthen the evidential basis of the claims.
In particular, the geometric residual consistency metric and the separability metric move beyond purely qualitative visualization and provide measurable criteria for assessing explanation plausibility and discriminative power.
This effort to quantify explanation quality represents a meaningful step toward reproducible evaluation in correspondence-based XAI.

Furthermore, the empirical validation is extensive.
The method is evaluated across more than thirty datasets spanning multiple wildlife and fine-grained re-identification domains, which is unusually comprehensive for explainability research.
Rather than relying on selective visual examples, the authors reported dataset-level statistics, cross-dataset trends, and correlation analyses, providing consistent evidence that the method performs strongly in texture-rich, pattern-based identity settings.
The inclusion of ablation analyses and intermediate-layer comparisons further demonstrates that the reported behavior is not incidental to a single architectural choice.

Finally, the manuscript does not restrict itself to favorable examples, which is important.
Failure cases were explicitly discussed, and experiments on ImageNet and Vision Transformer backbones were reported, even where the method’s effectiveness diminishes.
This transparency increases confidence that the conclusions are not overstated.
Moreover, the application to real-world animal re-identification workflows and engagement with domain experts provide practical grounding for the claims, connecting the technical contribution to realistic deployment scenarios.

Therefore, within the defined scope of fine-grained, correspondence-based explanation, the paper presented accurate and coherent empirical support for its central claims.
While certain limitations remain regarding metric assumptions and architectural generality, the evidence presented for the method’s effectiveness in its target regime is clear, structured, and convincingly demonstrated.

**Requested Changes:**

While the manuscript presents a technically coherent and empirically thorough contribution, several aspects would benefit from clarification and additional analysis.

First, the proposed method appears structurally dependent on identity being encoded in spatially localized, pattern-based visual cues.
As shown in the experiments on less-patterned species and general ImageNet classification, the separability between correct and incorrect pairs diminishes when similarity is driven by global or gestalt-level characteristics.
To strengthen the manuscript, the authors should explicitly formalize the scope of applicability of PAIR-X (e.g., pattern-dominant vs. global-structure-dominant regimes) and, if possible, provide a quantitative analysis linking performance to the degree of spatial localization of identity cues across datasets.
This would clarify whether the limitation reflects a scoped design choice rather than a weakness.

Second, the proposed geometric consistency metric relies on homography estimated via SuperPoint and LightGlue as a proxy for ground-truth alignment.
However, this reference geometry is itself model-derived and assumes a dominant planar transformation, which may not hold for articulated or non-rigid objects.
To address this concern, the authors should i) perform a sensitivity analysis using alternative correspondence pipelines (e.g., SIFT or other matchers), ii) evaluate the metric on synthetic image pairs with known geometric transformations to validate whether it tracks true alignment, and iii) report homography quality diagnostics such as inlier ratios, reprojection errors, and failure rates. These analyses would make the robustness and validity of the metric more transparent.

Third, although preliminary experiments on Vision Transformers are included, the evaluation relies on a CNN-oriented LRP adaptation and does not compare against transformer-specific relevance propagation frameworks.
In particular, established methods, such as [1] and [2], are not evaluated.
Since these methods are designed to handle attention mechanisms and residual connections in token-based architectures, incorporating them would clarify whether the observed transformer limitations arise from the relevance propagation choice or from deeper structural mismatch.
A controlled comparison on at least one transformer backbone would significantly strengthen claims of architectural generality.

## Reference
- [1] Chefer, Hila, Shir Gur, and Lior Wolf. "Transformer interpretability beyond attention visualization." Proceedings of the IEEE/CVF conference on computer vision and pattern recognition. 2021.
- [2] Achtibat, Reduan, et al. "AttnLRP: attention-aware layer-wise relevance propagation for transformers." Proceedings of the 41st International Conference on Machine Learning. 2024.

---

> ### Author Response · Authors · 2026-03-26
> **Response to Reviewer qGV4**
>
> Thank you for the thoughtful feedback! In response to the requested changes:
> - Thank you for the feedback on our technique’s performance on global/gestalt characteristics. We agree with this assessment; PAIR-X is best-suited for pattern-based tasks. We have added additional clarification on this point to Section 1.
> - We appreciate the suggestion on comparing to additional methods for generating a homography. In response, we have begun conducting experiments with additional methods for comparison, and will add them to the Supplementals shortly.
> - Thank you for highlighting additional transformer explainability baselines—we have added these references. PAIR-X offers fine-grained, pairwise explanations that may improve upon these methods, though it is currently limited by the LRP step. We hope future work will extend its effectiveness to transformers. Meanwhile, since CNNs remain widely used—especially in settings like animal re-ID with small, sparse datasets—we believe our method retains practical impact.

---

> > ### Comment · Reviewer_qGV4 · 2026-04-03
> >
> > After reviewing the revised supplementary material, I found that my concern about the validity of the geometric consistency metric remains unresolved.
> > The metric still depends on a homography estimated using SuperPoint and LightGlue, which cannot be regarded as true ground truth and is not clearly dependable for articulated or non-rigid objects.
> > Merely elaborating on the metric’s description is not enough.
> > Because this metric is central to the paper’s quantitative evaluation, the authors should support it with direct validation, for example by testing robustness across alternative matchers, evaluating it under synthetic transformations with known geometry, and reporting diagnostics on homography quality.
> > Without this kind of evidence, the reliability and interpretability of the metric are still uncertain.

---

> > > ### Author Response · Authors · 2026-04-11
> > >
> > > > After reviewing the revised supplementary material, I found that my concern about the validity of the geometric consistency metric remains unresolved. The metric still depends on a homography estimated using SuperPoint and LightGlue, which cannot be regarded as true ground truth and is not clearly dependable for articulated or non-rigid objects. Merely elaborating on the metric’s description is not enough. Because this metric is central to the paper’s quantitative evaluation, the authors should support it with direct validation, for example by testing robustness across alternative matchers, evaluating it under synthetic transformations with known geometry, and reporting diagnostics on homography quality. Without this kind of evidence, the reliability and interpretability of the metric are still uncertain.
> > >
> > > We apologize for the delay and have now added a sensitivity analysis of the PAIR-X metrics across two additional homography estimation methods (SIFT and ORB, in addition to the existing LightGlue results). This sensitivity analysis can be found in Supplementary Material F and Supplementary Table 4.
> > >
> > > Overall, we find that the PAIR-X metrics are largely invariant to the choice of homography estimation method. The Spearman rank correlations between the metrics based on LightGlue and SIFT / ORB are 0.8 or greater. We furthermore evaluated homography quality on a synthetic benchmark with known homographies and found that LightGlue performed best, recovering all homographies successfully with a reprojection error of 0.08 pixels.

---

### Review · Reviewer_B5Fa · 2026-03-06

**Summary Of Contributions:**

The authors are considering the problem of fine-grained feature attribution methods for explainability in paired-image object recognition, where the particular object is in fact a specific sample from a class of objects.  They propose to compose two ideas (feature subset matching and layerwise relevance propagation) that taken together will take a pair of images, and yield small sub-regions that are most implicated in a model classification over sameness-difference.

The authors make an extensive set of tests on public datasets, and also provide a user study of their tool (called PAIR-X).

**Audience:**

Yes

**Audience Explanation:**

I am not sure that there are many readers of TMLR that are concerned with the problem of re-identification or paired identification, but I am willing to give the authors the benefit of the doubt here.

**Broader Impact Concerns:**

There is not presently a broader impact statement, but I do think that one is warranted here.  One problematic use-case of re-identification here is surveillance.  The authors method could in theory be used to re-identify people based on surveillance footage, and this merits some consideration from them here.

**Claims And Evidence:**

No

**Claims Explanation:**

I have mixed conclusions here.  On one hand, Table 1 presents extensive experiments on a variety of different multi-species re-identification datasets.  Yet despite this, there are no comparisons with other methods on these same datasets; not on whether this method provides more accurate or effective features implicated in re-identification, or whether the method provides its paired feature attribution sets with greater computational efficiency.  So I find it quite difficult to evaluate whether there is a genuine advance presented in the work, because it does not seem like there are any experiments offered to evaluate this, even granted how difficult evaluating explainable AI methods.

**Requested Changes:**

## Critical changes

- Figure 3 is the key to summarizing how the method works, which as I understand from the text is something like:
1. compute embeddings for each image an input pair
2. for layer $l$, find a set of descriptors that match well between the representations of both images
3. use LRP to propagate those relevances down to pixel level

This is not easily understood from the figure, which has multiple arrows, headed in multiple directions.  The ordering could be made clearer.  For example, how does the stage the 2A - LRP relate to the features found in step 1?  It isn't very clear based on the diagram.  How is it that the features from step 1 get propagated back to pixel space by step 3 - LRP?  Surely step 2 alters these or selects from these,  but that does not come through in the figure.

- In section 3.2 equation (1), If I follow, the method flows as:
1. Find the closest matching activation sets
2. Use LRP to find the relevance values in those sets
3. Presuming that the sign of the R values must be the same for both images, multiply them

Is there a reason multiplication is preferred?  Would any other reduction preserve the match of relevance?  Where does cosine similarity, implicated in figure 3, come in?

- Section 3.3 proposes the use of an inverted residual mean, stating that quantifying explainability performance is difficult.  Yet there are many metrics and methods for feature based attribution, or which aim to quantify performance:
- AOPC
> Edin, J., Motzfeldt, A.G., Christensen, C.L., Ruotsalo, T., Maaløe, L., & Maistro, M. (2025). Normalized AOPC: Fixing Misleading Faithfulness Metrics for Feature Attribution Explainability. ACL 2025, pp. 1715–1730.
- SHAP
> Lundberg, S.M., & Lee, S.-I. (2017). A Unified Approach to Interpreting Model Predictions. Advances in Neural Information Processing Systems 30 (NeurIPS 2017), pp. 4765–4774.
- Integrated Gradients
> Sundararajan, M., Taly, A., & Yan, Q. (2017). Axiomatic Attribution for Deep Networks. Proceedings of ICML 2017, PMLR 70, pp. 3319–3328.
- Deep LIFT
> Shrikumar, A., Greenside, P., & Kundaje, A. (2017). Learning Important Features Through Propagating Activation Differences. Proceedings of ICML 2017, PMLR 70, pp. 3145–3153.

I think the authors should review these methods and explain how they either are unsuited to this use case, or explain what the difference is that motivates their heuristic proposed here in section 3.3.

## Major changes

- The **quantifiable metrics** point at the end of section 1 says
> Previous explainability approaches have often relied solely on manual, qualitative review to measure performance

Is this true of paired-image matching?  It is certainly not true of feature or instance based explainable AI work in general, and this difference should be made explicit.  Though I am not an expert in this specific area, I struggle to credit this statement.

- In section 3.1, the authors describe how the definition of keyoint locations does not fully capture the receptive fields of each neuron, and invoke LRP for visualization purposes.  This description of keypoint & location does not really convey that two images are being considered, both are decomposed into $l$ sets of features corresponding to the activations at the $l$-th layer of an arbitrary network intended to model images. Also I think that while LRP features briefly in section 2.2, it is given rather short shrift given its central importance to PAIR-X.  I think it merits a longer summary of its key points, and perhaps another reference to the original work here.

- The description in section 3.1 of how matching of descriptor sets is done leaves out much detail
> Given the complete sets of keypoints and descriptors, we perform brute-force matching with cross-checking

This strikes me as very odd.  There are ways that the authors could better align the sets of values to try and find near matching sets of values without resorting to brute-force search!  What might the cardinality of the matches be? For example, sorting and pairing would yield one such match with bounded complexity.

- In section 3.3, the authors describe how they measure explainability performance using classical techniques, describing the application of SuperPoint and LightGlue in series. I’m a bit confused about why both SuperPoint and LightGlue are being used here.  LightGlue is presented as a successor to SuperPoint, so is it really needed to make a homography?

- When I look at Figure 4, I’m trying to evaluate the matches but find the colour palate for the overlaid heatmap distracting; the emphasis is not so much on the matching regions, but rather dark colours of both ends of the palette are present, making it harder to focus on the high-value areas.  Maybe the authors could try incorporating a transformation of the alpha values for each, so that the more important match regions are more prominent?

Beyond the presentation of the matched poses or species, Figure 4 is a bit overwhelming.  It groups together several different concepts (oerformance on different classes of species, spurious correspondence identification, and an examination of pose variation.  I think these would be more effectively displayed as separate figures.

- Table 1 is well presented, but I think that the authors could establish that their heuristic score correlates with the GT by subsampling some rows.  By highlighing both successes and failures you can convince the reader without overwhelming them with so many entries.  The other entries could be moved to an appendix.

## Minor changes

- In section 3.1, the authors describe the activation matrix of layer $l$ as having dimensions $w×h×c$ that are downsampled.  Do they need to be downsampled?  Surely this is model dependent?  Or can they simply be the shape of the activations of the layer after $l$?

- At the end of section 3.2, the authors describe back-propagating from $M(A^{l}_{i,j}$.  “Backpropagate” in machine learning is usually understood as reverse mode differentiation.  But that is not what the authors mean here, they mean that the relevance value of this intermediate representation is collected through the conservation of relevance rules down to the pixel level.

- The **PAIR-X performs best  / less well on .....** sub-subsections are perhaps better suited to a discussion section, rather than Results.  I found it odd that they were grouped with the experimental results.

---

> ### Author Response · Authors · 2026-03-26
> **Response to Reviewer B5Fa**
>
> Thank you for all the detailed feedback! In response to each of the requested changes:
>
> ### Critical Changes
> - Thank you for your feedback on Figure 3. We have revised the figure to better convey the details of the method and to improve clarity, and would welcome any further suggestions.
> - Clarifications on equations
>   - Multiplication for combining relevances: Multiplying relevances favors balanced matches—penalizing uneven pairs (e.g., 3×1) and prioritizing pairs that are strong for both images (e.g., 2×2).
>   - Role of cosine similarity: The re-ID model is trained with cosine similarity, matching images via closest embeddings. Relevance is therefore computed with respect to the final cosine similarity, which serves as the starting point for backpropagation.
> - Why we don’t use existing metrics for feature-based attribution: while we agree that being able to apply existing quantitative metrics would be ideal, these metrics are designed for heatmap-style visualizations, and miss the key component of PAIR-X (pairwise spatial matches). Quantitatively comparing explanations that differ so much in composition becomes very difficult.
>
> ### Major changes
> - See above for explanation of why we don't use existing metrics for feature-based attribution
> - We have now clarified the descriptions of keypoints and descriptors, and added more in-depth explanation of LRP.
> - Why we use brute-force matching to find feature pairs: because the feature sets are small, this step is very cheap, especially relative to the steps that propagate through the model. We chose a simple approach because the size of the task did not warrant more complexity. But we’ll take this comment into account in case our approach ever changes in a way that makes the latency of this step more consequential!
> - Why we used both SuperPoint and LightGlue: we use SuperPoint to compute local descriptors, and LightGlue to match them. It’s true that LightGlue is a successor to SuperPoint; it builds on SuperGlue for faster matching.
> - Thank you for the comment on the visibility of images underneath the heatmap. We encountered the same challenge, but struggled to choose an opacity that balanced the utility of the heatmap with the utility of the image underneath. We added a clickable toggle in the PDF to hide the heatmap for clearer viewing. This works in some viewers (e.g., Firefox) but not others (e.g., Chrome or some macOS viewers), so we recommend using a compatible viewer.
> - Number of rows in Table 1: we appreciate the suggestion; however, we aim to provide a complete, transparent view across all datasets to assess consistency and generality, and thus prefer to retain the full table in the main text.
> ### Minor changes
> - Question about “downsampled” - by downsampled, we simply mean the reduced size of the feature matrix after layer l. We have clarified the wording in the manuscript.
> - Our use of the term ‘backpropagation’ - we use this term for consistency with existing work on LRP, which describes this step as “layerwise relevance backpropagation.” We have updated our language in the manuscript to be clearer.
> - Thanks for the suggestion to move our discussion of where PAIR-X performs well/poorly to Discussion - we’ve incorporated this change.
> - Impact statement: while the re-ID models explained by PAIR-X can be used for surveillance, our work does not develop these models but instead explains existing trained architectures. Thus, we did not include an impact statement.

---

> > ### Comment · Reviewer_B5Fa · 2026-03-27
> > **response to authors**
> >
> > > Why we use brute-force matching to find feature pairs: because the feature sets are small,
> >
> > That is true for the cases you tested, but I remain unconvinced that this is a good idea; what about for larger images or for multiple regions of interest?
> >
> > > Number of rows in Table 1: we appreciate the suggestion; however, we aim to provide a complete, transparent view across all datasets to assess consistency and generality, and thus prefer to retain the full table in the main text.
> >
> > I appreciate that you want to be transparent, but moving some of these to an appendix wouldn't change this.  I am not suggesting that they be omitted, I am suggesting they be better organized.
> >
> > While I raised a number of concerns in my review, the most pressing is the lack of comparison to other methods. The authors have a table where they showed their favoured metric aligned with another common metric, but they did not present these results (even for a subset of the datasets) against any other methods. How is a reader to know if this is any good or not?

---

> > > ### Author Response · Authors · 2026-04-03
> > > **Response to Reviewer B5Fa**
> > >
> > > >> Why we use brute-force matching to find feature pairs: because the feature sets are small,
> > >
> > > > That is true for the cases you tested, but I remain unconvinced that this is a good idea; what about for larger images or for multiple regions of interest?
> > >
> > > We agree that scalability is an important consideration. To assess this directly, we ran additional timing experiments. Within the PAIR-X pipeline, brute-force matching was not a bottleneck at the descriptor counts induced by the backbone layers we used: at Block 3 (28x28 = 784 descriptors), brute-force took 3.8 ms versus 13.3 ms for a k-d tree/FLANN-based matcher, and at the highest-resolution Block 1 (110x110 = 12,100 descriptors) the two were essentially tied (199.9 ms vs. 201.5 ms). In separate scaling experiments, the k-d tree based approach became advantageous only beyond roughly 14k descriptors, and was about 2x faster at 25.6k descriptors. To summarize, brute-force matching is sufficient for the experiments presented in this paper. However, we do agree that being able to handle prospective higher-resolution feature maps (as could, for example, be induced by higher-resolution images or new backbones) is important. Therefore, we are going to add the k-d tree/FLANN-based matcher as an option within the PAIR-X codebase.
> > >
> > > >> Number of rows in Table 1: we appreciate the suggestion; however, we aim to provide a complete, transparent view across all datasets to assess consistency and generality, and thus prefer to retain the full table in the main text.
> > >
> > > > I appreciate that you want to be transparent, but moving some of these to an appendix wouldn't change this. I am not suggesting that they be omitted, I am suggesting they be better organized.
> > >
> > > Based on your suggestions, we moved Table 1 into the Supplementary material and reduced the original table to five representative datasets. To better visualize the overall trends across datasets, we also added a scatterplot next to the summarized table, resulting in combined Figure 5 in the revised version. We welcome any further suggestions.
> > >
> > > > The authors have a table where they showed their favoured metric aligned with another common metric, but they did not present these results (even for a subset of the datasets) against any other methods. How is a reader to know if this is any good or not?
> > >
> > > We respectfully note that the manuscript does compare PAIR-X to multiple existing explainability methods in the main paper (Fig. 2) and Supplementary. What we do not provide is a single quantitative metric shared across all baselines, because the proposed PAIR-X metrics are method-specific: they assume explicit pairwise feature matching and therefore are not directly applicable to CAM-based explainability methods.

---

> > > > ### Comment · Reviewer_B5Fa · 2026-04-07
> > > > **Fair points**
> > > >
> > > > > We respectfully note that the manuscript does compare PAIR-X to multiple existing explainability methods in the main paper (Fig. 2) and Supplementary.
> > > >
> > > > I appreciate that the authors have made a good effort to compare PAIR-X to multiple explainability methods in the main paper.  Though it is somewhat limited in Fig. 2, it is more extensive in the supplemental.  I think it would help make the paper stronger if these were either referenced
> > > >
> > > > > What we do not provide is a single quantitative metric shared across all baselines, because the proposed PAIR-X metrics are method-specific: they assume explicit pairwise feature matching and therefore are not directly applicable to CAM-based explainability methods
> > > >
> > > > This is a point that I'd like to better understand specifically. PAIR-X operates on pairs of images: it both identifies relevant regions in each image, and matches these regions of interest.  A too-simple baseline using CAM-based methods could be first to use some CAM based method on each image to identify the regions of interest, and then to attempt to align them by the same matching heuristics.  An experiment like this should highlight more precisely how CAM-based methods fail to find the right fine grained features, or how the feature sets they find are too discordant to align manually, or how they fail to overlap with features identified by experts (by some orthogonal method), and therefore are deficient in the pairwise-feature matching metrics used for PAIR-X.

---

> > > ### Author Response · Authors · 2026-04-11
> > >
> > > > I appreciate that the authors have made a good effort to compare PAIR-X to multiple explainability methods in the main paper. Though it is somewhat limited in Fig. 2, it is more extensive in the supplemental. I think it would help make the paper stronger if these were either referenced
> > >
> > > Thank you for pointing this out! We have now referenced these additional results explicitly.
> > >
> > > > This is a point that I'd like to better understand specifically. PAIR-X operates on pairs of images: it both identifies relevant regions in each image, and matches these regions of interest. A too-simple baseline using CAM-based methods could be first to use some CAM based method on each image to identify the regions of interest, and then to attempt to align them by the same matching heuristics. An experiment like this should highlight more precisely how CAM-based methods fail to find the right fine grained features, or how the feature sets they find are too discordant to align manually, or how they fail to overlap with features identified by experts (by some orthogonal method), and therefore are deficient in the pairwise-feature matching metrics used for PAIR-X.
> > >
> > > We agree that the manuscript should more clearly distinguish qualitative baseline comparison from quantitative evaluation of PAIR-X itself. In the revised manuscript, we have made this distinction more explicit in Section 4 (Results, first paragraph). In particular, we have clarified that Fig. 2 and Supplementary Fig. 10 provide the cross-method comparison to diverse baseline methods, whereas the metrics in Section 3.3 are method-specific PAIR-X metrics intended to evaluate the plausibility of PAIR-X explanations across datasets and settings.
> > >
> > > The reason we do not report a single shared quantitative score against CAM-based methods is that the PAIR-X metrics are defined on explicit pairwise correspondences and relevance-weighted matched regions. CAM-based methods do not natively produce such correspondences, therefore, introducing a "CAM + matching heuristics" baseline would therefore require additional design choices for region extraction, thresholding, coordinate definition, and matching, effectively introducing a new hybrid method rather than evaluating an existing baseline under a common metric.
> > >
> > > We agree that such hybrid baselines could be interesting to study in future work. However, our main goal is to compare PAIR-X qualitatively against existing explainability methods, and to quantify PAIR-X in a way that helps users determine whether it produces meaningful and practically useful explanations for their particular dataset and use-case.

---

### Review · Reviewer_Rdqi · 2026-03-12

**Summary Of Contributions:**

This submission proposes a new approach to use intermediate model activations and scores from backpropagation to generate visual explanations. To better evaluate explainability performance, this work also proposes new evaluation metrics. Qualitatively, the proposed method has been evaluated and validated on a wide range of dataset. Quantitatively, the proposed method has been have demonstrated its advantages in interpretabilities.

**Audience:**

No

**Audience Explanation:**

I am negative on this section. First, the proposed method has not shown significances compared to other methods in this community, CAM-based methods have been widely used with significant advantages, the results shown in this paper have not been superior. The key component of side-side comparisons on the same metrics seems to be missing. Second, the method is not significant to the community, especially we have seen prior work fully leverage intermediate activations with a series of downstreaming tasks [1]. This paper has not shown downstreaming evaluations which are comparable to CAM-based methods, like unsupervised learning, segmentation and detection, etc. Overall, I think this work is less attractive to the community.


[1]. Sharpen Focus: Learning with Attention Separability and Consistency. Wang et al. 2019.

**Broader Impact Concerns:**

I did not find significant concerns for this section.

**Claims And Evidence:**

Yes

**Claims Explanation:**

This work claims three major component:

1. It proposes PAIR-X to extract correspondences between image pairs with explicit visualizations. Qualitative results have demonstrated its effectiveness.

2. This work highlight that it provides explanations in the original resolution which maintains the most details and the visualizations have help validate it with comparisons with other work.

3. This work propose new metrics to effectively measure explanations, section 3.3 has provided quantitative results, comparisons and discussions.

Overall, I think this work has focused on its claims with both qualitative and quantitative evidence.

**Requested Changes:**

1. Figure 3 is not clear enough, it is confusing to read 1, 2A, 2B, 3, authors can use different color to indicate different modules/blocks with better flows.

2. Since CAM-based methods have been widely used in the community, have authors evaluate the proposed method on some downstreaming tasks, like segmentation, detection, OOD, etc?

3. Quantitatively, it will also be helpful to compare with CAM-based method on the proposed metrics as well as existing metrics?

4. Are there any results from expert interviews? It is quite popular to provide results from human eval, raters, etc?

5. CAM-based methods have been extremely useful in both academia and industries since many of them donot require training and efficacy. As mentioned earlier, combing intermediate activations has also been studied before [1]. I think this work lacks deeper discussions between the proposed method and CAM-based method.

[1]. Sharpen Focus: Learning with Attention Separability and Consistency. Wang et al. 2019.

---

> ### Author Response · Authors · 2026-03-26
> **Response to Reviewer Rdqi**
>
> Thank you for your feedback! In response to each of the requested changes:
> 1. Thank you for your helpful suggestions on Figure 3. We’ve made some revisions to the figure, and would appreciate any additional feedback you may have.
> 2. We agree that applying PAIR-X to downstream tasks would be an interesting direction for future work. However, as the primary goal of the method is explainability, we consider this beyond the scope of the present paper.
> 3. Unfortunately, the proposed metrics are specific to the pairwise nature of PAIR-X explanations: they are designed to evaluate the quality of matches between images and aren’t applicable to heatmap-based methods such as CAM. We agree that additional quantitative comparisons would be valuable; however, differences in the structure of these explanation types (heatmaps versus explicit correspondences between image regions) make such comparisons challenging.
> 4. We do include feedback from domain experts; please see Section 5.2.1 for details.
> 5. We appreciate the feedback regarding the utility of PAIR-X relative to existing explainability techniques. However, we still think that PAIR-X is novel and improves meaningfully over CAM-based methods. PAIR-X was developed to address limitations of approaches like CAM in fine-grained, pairwise tasks such as animal re-ID while remaining training-free. In particular, its pairwise formulation enables rapid visual alignment between image pairs, and its fine-grained correspondences capture details (e.g., spot patterns) that are often not resolved by coarse heatmaps. We have also seen encouraging interest from practitioners in this space; for example, WildMe (a nonprofit focused on animal re-ID) has developed a fork of our repository for use in field applications.

---

> > ### Author Response · Authors · 2026-04-11
> >
> > Dear reviewer, did our response address your concerns? Please feel free to let us know if you have any further questions.

---

### Author Response · Authors · 2026-03-26

Thank you to all the reviewers for their detailed and helpful feedback! We have submitted a revised version of the manuscript and believe that we have addressed the requested changes; we have also provided more detailed responses to each reviewer in the threads below. We would welcome any additional feedback.

---

### Decision · Action_Editor_h3Vr · 2026-04-19

**Recommendation:** Reject

**Audience:**

Yes

**Audience Explanation:**

While the answer would be 'yes', there were some concerns on that aspect as well voiced by one of the reviewers:
- While the pipeline focuses on pairwise data explanations, it may limit the appeal of the paper to the audience.

I suggest the authors add some justification to the introduction: while the main goal might be animal re-identification, emphasising the wider applicability would strengthen the audience criterion.

**Claims And Evidence:**

No

**Claims Explanation:**

The reviewers are equivocal that although the work appears promising and while the presentation has been improved during the discussion period, there are concerns that not all suggestions by the reviewers have been fully addressed to the extent to warrant acceptance.

These concerns are mainly centred around the lack of sufficiently strong and appropriate baseline comparisons (all reviewers).

I would like to highlight the discussion with Reviewer B5Fa, which is concerned about setting up the quantitative baseline for the method and therefore about demonstrating that "by improving interpretability, PAIR-X enables
humans to better distinguish correct and incorrect matches" in a way that uses quantitative arguments:

R B5FA:

*"This is a point that I'd like to better understand specifically. PAIR-X operates on pairs of images: it both identifies relevant regions in each image, and matches these regions of interest. A too-simple baseline using CAM-based methods could be first to use some CAM based method on each image to identify the regions of interest, and then to attempt to align them by the same matching heuristics. An experiment like this should highlight more precisely how CAM-based methods fail to find the right fine grained features, or how the feature sets they find are too discordant to align manually, or how they fail to overlap with features identified by experts (by some orthogonal method), and therefore are deficient in the pairwise-feature matching metrics used for PAIR-X."*

Authors:

*"The reason we do not report a single shared quantitative score against CAM-based methods is that the PAIR-X metrics are defined on explicit pairwise correspondences and relevance-weighted matched regions. CAM-based methods do not natively produce such correspondences, therefore, introducing a "CAM + matching heuristics" baseline would therefore require additional design choices for region extraction, thresholding, coordinate definition, and matching, effectively introducing a new hybrid method rather than evaluating an existing baseline under a common metric.*

*We agree that such hybrid baselines could be interesting to study in future work. However, our main goal is to compare PAIR-X qualitatively against existing explainability methods, and to quantify PAIR-X in a way that helps users determine whether it produces meaningful and practically useful explanations for their particular dataset and use-case."*

I think such a study would exactly answer the following question: why is it necessary to propose this method? Why doesn't the simplest possible extension of CAM and matching solve this problem? I would expect that such a simplest possible extension does not solve it and this method is probably going to significantly improve upon it, however, having a quantitative comparison would be an important way to show it.

The updated results ("Suppl. Sec. E for results on an additional model, and Suppl. Sec. H for a preliminary expansion to
transformers") address critical questions of evaluation on additional metric models, and deserve to be in the main text; now it seems like it attaches the additional results to the paper without incorporating them into the text and referencing them with the joint conclusion. I would personally think that while the paper improvements took a reasonable direction, I would suggest to revise it thoroughly incorporating the comments and showing the advantage of the method, take into account the comments from the reviewers and then resubmit. Therefore, my recommendation is resubmission of the major revision.

**Resubmission Of Major Revision:**

The authors may consider submitting a major revision at a later time.